# COVID-19 and Liquid Homeostasis in the Lung—A Perspective through the Epithelial Sodium Channel (ENaC) Lens

**DOI:** 10.3390/cells11111801

**Published:** 2022-05-31

**Authors:** Emily F. Brown, Tamapuretu Mitaera, Martin Fronius

**Affiliations:** 1Department of Physiology, University of Otago, Dunedin 9054, New Zealand; emily.brown@otago.ac.nz (E.F.B.); mitta940@student.otago.ac.nz (T.M.); 2HeartOtago, University of Otago, Dunedin, New Zealand; 3Healthy Hearts for Aotearoa New Zealand, Centre of Research Excellence, New Zealand; 4Maurice Wilkins Centre for Molecular Discovery, Centre of Research Excellence, New Zealand

**Keywords:** COVID-19, SARS-CoV-2, epithelial Na^+^ channel, proteases, proteolytic cleavage, alveolar liquid, airway liquid, transepithelial ion transport

## Abstract

Infections with a new corona virus in 2019 lead to the definition of a new disease known as Corona Virus Disease 2019 (COVID-19). The sever cases of COVID-19 and the main cause of death due to virus infection are attributed to respiratory distress. This is associated with the formation of pulmonary oedema that impairs blood oxygenation and hypoxemia as main symptoms of respiratory distress. An important player for the maintenance of a defined liquid environment in lungs needed for normal lung function is the epithelial sodium channel (ENaC). The present article reviews the implications of SARS-CoV-2 infections from the perspective of impaired function of ENaC. The rationale for this perspective is derived from the recognition that viral spike protein and ENaC share a common proteolytic cleavage site. This cleavage site is utilized by the protease furin, that is essential for ENaC activity. Furin cleavage of spike ‘activates’ the virus protein to enable binding to host cell membrane receptors and initiate cell infection. Based on the importance of proteolytic cleavage for ENaC function and activation of spike, it seems feasible to assume that virus infections are associated with impaired ENaC activity. This is further supported by symptoms of COVID-19 that are reminiscent of impaired ENaC function in the respiratory tract.

## 1. Pulmonary Liquid Homeostasis from an Evolutionary Perspective

Ion transport processes due to the function of the pulmonary epithelial are crucial for the function of the lung. It is important to consider the relevance of pulmonary ion transport in the background of water preservation related to the water-to-land transition for habitat expansion as part of an evolutionary processes.

Living cells require a liquid environment to be able to maintain intracellular homeostasis. The exposure to air, due to water evaporation, provides considerable challenges for intracellular homeostasis. Thus, the transition to land was accompanied with the emergence of body surfaces that limit the loss of water and provide mechanical protection against the physical environment. The human skin is an obvious example of such a body surface. It is a layer that limits water evaporation and provides protection against the elements. Both functions depend on the outer layer of this barrier, the Stratum Corneum. However, it is important to emphasize that the human Stratum Corneum is composed of dead cells that are shed off from epithelial cell layers underneath. These dead cells composing the Stratum Corneum do not need a liquid environment.

Limiting water evaporation is one challenge that was derived from the water-to-land transition. Another challenge is the ability to maintain gases’ exchange via diffusion from air. This per se does not rely on lungs. Some amphibian species do not have lungs and utilize their skin for gas exchange by diffusion [1]. Relying on the skin for gas exchange provides considerable limitations to body size and metabolic activity [2]. Importantly, it depends on aquatic habitats, because the skin needs to be kept moist, meaning that amphibians are still bound to live in or close by freshwater habitats. The amphibian skin possesses a sophisticated ion transport system that allows them to keep the skin moist when exposed to air, but, at the same time, it is designed to limit water loss due to the reabsorption of Na^+^ and Cl^−^ ions, providing the driving force for water movement [3].

In contrast, the human skin is a considerable barrier for gas diffusion and unsuitable to provide sufficient gas exchange for survival. Thus, specialized body surface areas emerged for gas exchange from air—the lungs. Lungs are still body surfaces [4]. They are lined by an epithelial layer (Figure 1), and this layer faces the same challenges as other body surface areas that are exposed to air: they (1) maintain a liquid environment to facilitate cell homeostasis and (2) provide a certain degree of protection against the environment. Both aspects are accomplished by the invagination of the lungs with its approx. 100 m^2^ surface area delicately folded to fit into the chests. Due to the requirement for the lung surface to be permeable for breathing gases, the barrier in the gas exchanging regions—the alveoli—is extremely thin [5,6]. This is achieved by a single-cell-layered epithelium of cells (alveolar type 1 and type 2 cells) that form the alveoli. These cells are living cells, and they need a liquid environment to be able to survive. Reminiscent to the frog skin as a respiratory organ, the lung surface exposed to ‘air’ is covered by a thin film of liquid and additionally is surrounded by a water-saturated atmosphere [7]. In the alveoli, the thickness of the liquid layer needs to be finely balanced for two main reasons: (1) to limit water loss from the cell’s cytoplasm to maintain cellular homeostasis and cell function; and (2) excess liquid will impair gas diffusion because it increases the diffusion distance for the breathing gases. This implies that lung surfaces such as the alveolar epithelium requires mechanisms that maintain this wet environment.

## 2. Sodium Transport Is Essential for Pulmonary Liquid Homeostasis

A common principle for the movement of water across barriers (e.g., cell membranes or epithelial structures) is the generation of osmotic gradients. Due to its dipole properties, water follows the movement of ions. Thus, vectorial ion transport provides the driving forces for either secretion or absorption of water and is essential for the regulation and maintenance of water homeostasis. Water secretion is driven by transcellular Cl^−^ secretion, where water follows the osmotic gradient and water absorption is driven by transcellular Na^+^ absorption (Figure 2) [8]. Both Cl^−^ secretion and Na^+^ absorption are essential for normal lung function, as indicated by underlying pathologies rooted in impaired ion transport [9]. The following section briefly highlights the role of Na^+^ absorption for liquid homeostasis in the alveolar and airway region of the lungs.

### 2.1. Sodium Transport across the Alveolar Epithelium

An essential function of the alveolar epithelium is to balance the liquid content in the alveolar space [10]. This involves transepithelial ion transport processes and is required for facilitating gas exchange. This is highlighted by observations that impaired pulmonary ion transport is lethal. Transgenic mice lacking the alpha (α) subunit of the epithelial Na^+^ channel (ENaC) die within hours after birth because they are unable to reabsorb the liquid from the airspace of their lungs [11]. This implies that the reabsorption of water relies on the function of ENaC and that Na^+^ reabsorption provides the driving force for water reabsorption. Impairment of Na^+^ reabsorption impairs alveolar liquid homeostasis and subsequently gas diffusion [4,12]. The association between impaired liquid balance and impaired gas exchange is further supported by observations that respiratory distress and hypoxemia are commonly associated with the formation of pulmonary oedema [13].

### 2.2. Sodium Transport across the Airway Epithelium

In the airways, the best-known function of epithelial ion transport accomplished by the airway epithelia is to maintain the thickness and viscosity of the liquid layer covering the epithelium to facilitate mucociliary clearance, an essential function of the innate immune system [14,15]. This role is highlighted by cystic fibrosis lung disease caused by a mutated Cl^−^ channel (cystic fibrosis transmembrane conductance regulator, CFTR) [16,17]. The lack of Cl^−^ secretion leads to water depletion from the liquid layer and impairs the beating of cilia [18]. Similar to the alveolar epithelium, ENaC also plays an important role for the homeostasis of the liquid layer covering the airways. Astonishingly, gain-of-function mutations of ENaC have been identified to cause a cystic-fibrosis-like phenotype in patients with functional CFTR [19,20]. The gain of function of ENaC will have similar consequences as the loss of function of CFTR—depletion of water from the airway surface layer and impaired mucociliary clearance. Similar observations were also reported from a transgenic mouse model. Here, the overexpression of the β ENaC subunit in the airways was observed to cause a cystic-fibrosis-like phenotype in the lungs [21]. The importance of Na^+^ absorption to maintain water balance in the airways is further emphasized by studies showing that transgenic mice with impaired CFTR function did not develop lung disease as anticipated [22] and sparked the quest for alternative transgenic animal models [23].

In summary, Na^+^ reabsorption via ENaC in both the alveolar epithelium and the airways is essential for maintaining a certain liquid environment to support the respective function of the alveoli and the airways. It is also worthwhile to emphasize the similarities of Na^+^ mediated water homeostasis in pulmonary epithelia and in the frog-skin epithelium.

## 3. ENaC Regulation by Proteases

As indicated, ENaC plays an essential role for liquid homeostasis in the lungs. ENaC proteins (α ENaC) were identified by cloning from a rat colon [24] and human lungs [25]. Two additional subunits, named β and γ ENaC, were identified, and it was observed that the combination of all three subunits (α, β, and γ) provided high amiloride-sensitive currents when co-expressed in Xenopus oocytes [26]. A fourth subunit, δ ENaC, was identified in human tissues [27]. This subunit is also expressed in the lung, but its role remains unknown [28]. Recent evidence suggests that the canonical channel formed by α, β, and γ ENaC subunits is a trimer, where subunits assemble in a counterclockwise manner [29]. Because ENaC is a key regulator for electrolyte and water homeostasis [30], its activity is regulated via several mechanisms. However, the present article will discuss only one mechanism that is involved in regulating ENaC’s open probability and relates to SARS-CoV-2 infections—the regulation via proteases and the proteolytical cleavage of ENaC.

This mechanism was identified by the discovery of a protease that was named CAP1 (channel activating protease 1). The co-expression of ENaC and CAP1 in *Xenopus* oocytes resulted in elevated ENaC currents, indicating that proteolytic cleavage of ENaC by CAP1 activated the channel [31]. Further studies revealed that uncleaved channels have a low open probability, and the proteolytic cleavage increases the open probability of the channel [32,33]. Regulation of ENaC activity via proteolytic cleavage is complex because multiple proteolytic cleavage sites have been identified in the α and γ ENaC subunits [34,35]. Furthermore, the cleavage sites in γ ENaC can be cleaved by more than one protease [36]. Moreover, to add more complexity, there are intracellular proteases (e.g., furin) and extracellular proteases (e.g., CAP1), with some of the latter being either membrane-anchored or secreted to be freely available in the extracellular space [37].

The recent understanding is that fully activated ENaC comprises an α subunit that is cleaved twice by the protease furin during the transition through the trans-Golgi network and a γ subunit that is also cleaved once by furin in the trans-Golgi network and another protease at the cell membrane (e.g., CAP1) [37,38,39]. The proteolytic cleavage of the subunits can be sequential and occur in various locations (intra- and extracellularly), and this is suggested to provide opportunities for a graduate activation and regulation [39]. Resolving the ENaC structure by cryo-electron microscopy identified a domain within each subunit that was named GRIP (Gating Release of Inhibition by Proteolysis). The proteolytic cleavage of both sites within the α and the γ ENaC subunit is suggested to release inhibitory peptides [40]. This induces a conformational change and facilitates channel gating [29].

### Regulation of ENaC in the Lung by Proteases

Proteolytic cleavage was also identified to regulate ENaC activity in the lung. Research in primary nasal epithelial cells [41] and primary bronchiolar epithelial cells [42] identified endogenous proteases that were involved in activating ENaC. The importance of this mechanism was further supported by a study of Planes and co-worker that utilized transgenic animals. In a transgenic model lacking CAP1 in alveolar epithelial cells, ~50% reduced amiloride sensitive alveolar liquid clearance was observed. It might be noted that the lack of CAP1 per se did not result in a pathological phenotype [43]. However, these animals had an impaired ability to reabsorb liquid from the alveolar airspace in situations of experimentally induced pulmonary oedema [43]. This highlights the importance of CAP1 for ENaC regulation in the lung in situations associated with liquid accumulation in the alveolar region. This finding is in agreement with in vitro studies that found that proper ENaC function in the lung relies on the activity of multiple proteases and not just CAP1 [38,44]. Nevertheless, the understanding of how certain proteases regulate ENaC function in vivo remains mysterious, as indicated by the study of Donaldson and colleagues [41]. In contrast to the majority of reports, Donaldson and colleagues identified a decreased ENaC activity in the presence of a protease named TMPRSS2 (Transmembrane Protease Serine 2).

## 4. SARS-CoV-2 and Corona Virus Disease 2019 (COVID-19)

Given their relative exposition to the environment and their delicate structure, it is no surprise that the lung epithelia are susceptible to infections by pathogens, and particularly viruses that are transmitted via air droplets, including corona viruses [45]. The start of the recent COVID-19 pandemic was accompanied by reports of a new type of corona virus [46]. This new virus did cause similar respiratory distress, as observed with previous viruses such as SARS-CoV (Severe Acute Respiratory Syndrome Coronavirus) and MERS-CoV (Middle East Respiratory Syndrome Coronavirus) [47,48]. A major reason for respiratory distress because of viral infections is the accumulation of liquid that impairs gas exchange and causes hypoxemia. Thus, the observed pulmonary oedema is indicating impaired water homeostasis in the lung.

The reasons for impaired water homeostasis can be manyfold and as simple as pure leakiness of the air–blood barrier due to impaired barrier function caused by dying/infected epithelial cells and the inability to ‘heal’. Another implication which can have similar consequences is impaired ion transport by lung epithelia and, more specifically, impaired Na^+^ reabsorption, as implied by the observations form the transgenic animal lacking α ENaC.

Sequence alignments comparing the SARS-CoV-2 genome with SARS-CoV and other corona viruses identified unique changes in the genome. The SARS-CoV-2 genome encoding the spike protein only shares ~40% amino acid identity with SARS-CoV [46]. The spike protein is a glycoprotein and forms a trimeric receptor protein complex at the surface of corona viruses (Figure 3) [49,50]. It consists of two main domains (S1 and S2) that are separated by a cleavage site. Cleavage of the spike protein via host proteases is essential for initiating the binding of the virus to host cells and the infection [51]. Cleaved spike proteins undergo extensive conformational change, and this enables binding to the angiotensin converting enzyme 2 (ACE2) receptor to initiate membrane fusion [52]. The subsequent uptake of the viral proteins into the host cell activates the viral replication machinery.

The proteolytic cleavage of spike is a key mechanism for ‘activating’ the virus. Thus, proteases emerge as major targets for antiviral treatment [53]. It is interesting to note that the cleavage of spike, similar to the cleavage of ENaC, is a key step to initiate conformational change and ‘activate’ the respective protein.

In contrast to previous corona virus outbreaks, SARS-CoV-2 was more infectious compared with other SARS-CoVs and MERS-CoVs [54]. Early on, it was suggested that a new proteolytic cleavage sequence in the spike protein could be a reason for the high infectiousness [55]. This new cleavage site is localized between the interface of the S1 and S2 subdomains of the spike protein (Figure 3). The addition of amino acids resulted in a sequence _683_RRARSVAS_690_ [55], representing a multi basic amino acid motif that is a suitable recognition site for the protease furin [56].

## 5. α-ENaC and SARS-CoV-2 Spike Protein Have Identical Proteolytic Cleavage Site

Anand and colleagues have used this new viral furin sequence (RRARSVAS) for an in silico study to perform sequence alignments against human proteins [57]. Amongst the 23,000 proteins they included in their analyses, they reported one human protein that shared the exact same sequence. The identified protein is the human α-ENaC subunit [57]. In ENaC, this sequence (_201_RRARSVAS_208_) is one of the cleavage sites for the protease furin (Figure 4). This proteolytic cleavage site in the spike protein is targeted by furin during packaging of the virus inside host cells [58]. This enables ‘pre-activation’ of the virus and allows binding to host ACE2 receptors [58]. The pre-activation by furin is considered to be an important new feature of SARS-CoV-2 in comparison to SARS-CoV because viruses carrying the furin cleaved spike can infect other cells without the necessity to be activated by extracellular proteases, as is known for SARS-CoV [56]. If and how the occupation of furin by SARS-CoV-2 spike protein affects ENaC function is unknown. However, given the importance of furin for ENaC activation, it seems likely that spike could interfere with ENaC cleavage in infected cells whose function relies on ENaC mediated Na^+^ uptake. This also provides a new and interesting perspective to explain some of the symptoms associated with SARS-CoV-2 infections in COVID-19 through an ENaC lens and compare symptoms with known loss/gain of functions of ENaC.

## 6. Are ENaC and Spike Targeted by the Same Proteases?

Impaired ENaC function has been associated with viral infections such as SARS-CoV [59] and influenza [60,61,62]. The role of proteases for this association has recently been discussed [63,64,65], but experimental evidence is missing.

Besides furin, other extracellular host proteases have been identified to be important for the cleavage of the spike protein. In addition to the furin site at the S1/S2 interface that facilitates cell attachment, SARS-CoV-2 spike possesses another cleavage site termed S2′. Cleavage of this site facilitates membrane fusion and cell entry of the virus cell entry of the virus [56]. The importance of this site for cell infection and the protease that cleaves the site is known from previous work studying SARS-CoV. The protease targeting the S2′ site is TMPRSS2 (TransMembrane PRotease Serine 2) [66]. In SARS-CoV-2, the S2′ site can also be cleaved by TMPRSS2 [52]. In addition to TMPRSS2, another protease belonging to the same protein family, TMPRSS4, has been identified to facilitate cleavage of SARS-CoV-2 spike and infections of host cells [67]. Although evidence for the proteolytic cleavage of ENaC by TMPRSS2 is missing, the protease does reduce ENaC protein (and thus function) via an unknown mechanism [41]. Moreover, there is evidence that TMPRSS4 can cleave the γ ENaC subunit in vitro [68]. However, deletion of TMPRSS4 does not seem to be involved in regulating ENaC activity in the kidney or colon [69].

**Figure 4 cells-11-01801-f004:**
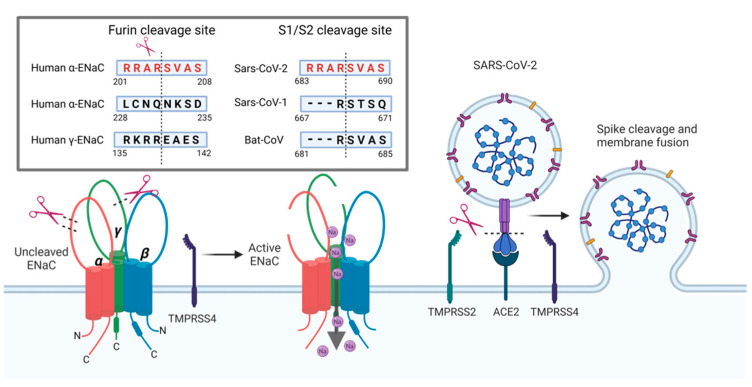
Furin cleavage sites for α- and γ-ENaC subunits and SARS-CoV-2 spike protein. Furin cleaves the α-subunit twice as well as the γ-subunit to fully activate the channel. Intriguingly, the SARS-CoV-2 spike protein has an identical 8 amino acid (aa) sequence as one of the furin cleavage sites of α-ENaC. Cleavage at this site increases SARS-CoV-2 spike’s affinity to the ACE2 receptor compared to SARS-CoV spike. SARS-CoV-2 spike protein is further cleaved by TMPRSS2 and TMPRSS4 to facilitate viral entry to the host cell. TMPRSS4 can also cleave γ-ENaC at a cleavage site that is distinct from the furin cleavage site. The occupation of host proteases by SARS-CoV-2 may reduce proteolytic cleavage of α- and γ-ENaC, which will impair Na^+^ transport. Furin sequences: human α-ENaC, aa201-208, SARS-CoV-2, aa683-690, SARS-CoV, aa667-671 and Bat-CoV, aa681-685 published in reference [57]; human α-ENaC, aa228-235 published in reference [36]; and human γ-ENaC, aa135-142 published in reference [68] (created by BioRender.com).

Nevertheless, the striking similarity of the furin cleavage sequence between the initial SARS-CoV-2 spike protein and α-ENaC, and the evidence that TMPRSS2/4 can interfere with both, spike and ENaC proteins, strongly indicates a potential interference of ENaC cleavage in the presence of viral spike protein by impaired proteolytic cleavage. Furthermore, ENaC and TMPRSS protease co-expression was reported in various tissues, including the pulmonary tract [57].

Although clear experimental evidence is missing of whether ENaC and spike protein are processed by identical host proteases, proteases play a key role for the infectiousness of SARS-CoV-2 and for ENaC activity. Identifying ‘one’ protease that is crucial for this process may be impossible because the proteolytic cleavage system displays a considerable redundancy. Cleavage motifs within one protein can be recognized by multiple protease [70]. Cleavage of the same sequence may therefore occur in cells expressing different proteases. An example for the proteolytic redundancy is the proteolytic cleavage of γ ENaC. For one of its cleavage sites, >10 proteases have been identified that can cleave this subunit [71].

## 7. COVID-19 Respiratory Symptoms through an ENaC Lens

It is feasible to assume that cells that express ENaC and its proteolytic cleavage machinery are susceptible to viral infections because the same proteases would be able to cleave the spike protein. Therefore, host proteases needed for ENaC activation facilitate binding of the virus to the AEC2 receptor and initiate the cell infection. Moreover, ENaC subunits (and presumably the proteases needed for regulating its activity) are expressed along the entire respiratory tract, including the tongue [72,73,74], the nasal epithelium [75], the airways [76,77], and the alveolar epithelial cells [78,79,80].

The infection of cells by SARS-CoV-2 in the respiratory tract can impair underlying cell function and results in reported symptoms such as loss of taste/smell, runny nose, cough, and pulmonary oedema depending on the region and severity of the viral infection. In theory, these symptoms can also be explained by impaired ENaC function in the respective cells.

ENaC expression in taste receptor cells of the tongue is important for salt tasting. Here, ENaC is considered to depolarize taste cells in the presence of Na^+^ to initiate the perception of salt [73]. Thus, impaired ENaC activity will impair salt tasting.

In the airways, a reduced ENaC activity was described in patients with pseudohypoaldosterism. In these patients, this was associated with the build-up of liquid in the airways and impair mucociliary clearance, leading to coughing and sneezing [81,82]. Similarly, impaired ENaC activity in the nasal epithelium is linked to chronic rhinitis (runny nose) [30].

Finally, impaired ENaC function on the alveolar region will impair liquid handling and can result in liquid build-up in the alveolar region to cause alveolar oedema and respiratory distress [11,83].

## 8. Does SARS-CoV-2 Overwhelm the Proteolytic System and Prevent ENaC Function?

Impaired ENaC function may be caused by an overwhelmed proteolytic cleavage machinery. Thus, the exposure of host proteases to SARS-CoV-2 spike protein may interfere with the ability to preserve ENaC function and was also discussed previously [64]. Impaired ENaC function could be a major contributor for the formation of pulmonary oedema, as reported in patients with severe COVID-19.

This aspect is supported by the reported clinical manifestation of pulmonary oedema and ARDS (acute respiratory distress syndrome) caused by SARS-CoV-2 virus infection. ARDS and pulmonary oedema formation in COVID-19 patients is described to be distinct from ‘classic’ ARDS. In COVID-19, the progression of the disease to ARDS is defined as a ‘severe leak’, but without the elevated pulmonary pressure, which is known for high-altitude pulmonary oedema and ‘classic’ ARDS [84]. Similarly, another study reported an increased extravascular lung water index and elevated pulmonary vascular permeability index in COVID-19 ARDS in comparison to ‘classic’ ARDS. This was accompanied by a lower hemodynamic severity in COVID-19 ARDS compared to ‘classic’ ARDS, indicating a lower impairment of the pulmonary vasculature [85]. This suggests that pulmonary oedema formation and ARDS in COVID-19 may be caused by a different mechanism.

A putative explanation for the differences between COVID-19 ARDS and ‘classic’ ARDS could be that oedema formation leading to respiratory distress and hypoxemia in COVID-19 is caused—at least partially—by impaired ion transport processes across the alveolar epithelium.

## 9. Summary

Considering the importance of pulmonary Na^+^ reabsorption through ENaC for maintaining gas exchange in the lung, it should not be surprising that interference with these processes can have serious consequences. The recent pandemic caused by the highly transmissible SARS-CoV-2 virus can be explained by the new furin cleavage site. The similarity of the initial SARS-CoV-2 furin cleavage site with a furin cleavage site in α ENaC is astonishing and provides a unique opportunity to compare COVID-19 symptoms with symptoms associated with impaired ENaC function. A suitable mechanism to explain the similarity of the symptoms is the impaired activation of ENaC due to impaired proteolytic cleavage. This can be caused by the virus overwhelming the proteolytic cleavage machinery of cells that express ENaC and, therefore, preventing proteolytic cleavage of ENaC. Thus, it might be speculated that improving ENaC function can be achieved by strategies that increase proteolytic cleavage. Improving ENaC cleavage in pulmonary epithelia could therefore be a suitable strategy to treat pulmonary oedema formation in COVID-19 patients with respiratory distress. However, this relies on an approach that aims to increase the protease activity of proteases that have a high affinity for ENaC but are less effective in cleaving the SARS-CoV-2 spike protein. This may include proteases that can effectively cleave the non-furin cleavage site of γ ENaC but have a low affinity for SARS-CoV-2 spike protein.

## Figures and Tables

**Figure 1 cells-11-01801-f001:**
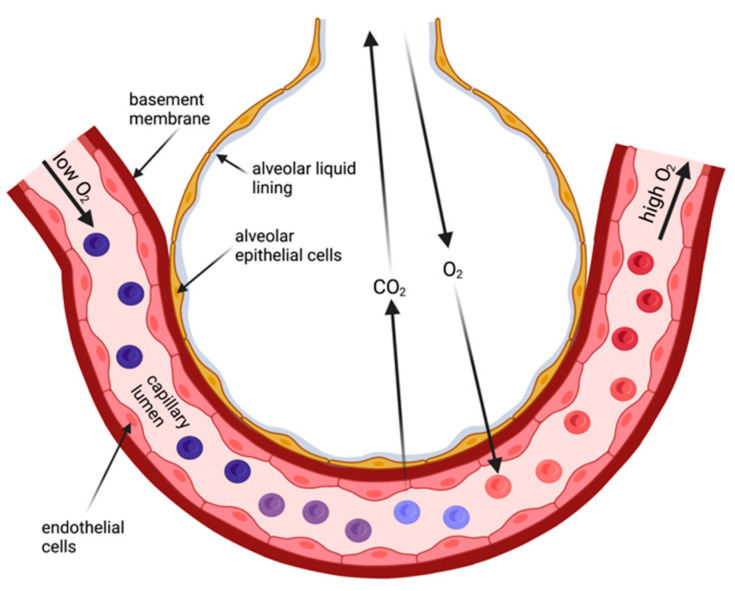
Schematic drawing of the alveolar region forming the air–blood barrier. The air–blood barrier comprises a liquid layer covering the alveolar epithelium, the alveolar epithelium, the basement membrane, and the endothelial cell layer. CO_2_ and O_2_ cross the air–blood barrier by diffusion (created with BioRender.com).

**Figure 2 cells-11-01801-f002:**
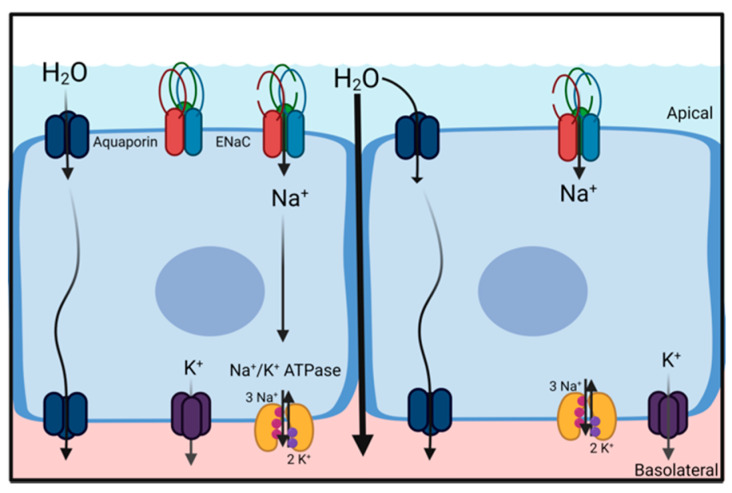
Principal mechanism for Na^+^ driven water absorption in absorptive epithelia, including alveolar and airway epithelia. Na^+^ absorption is facilitated by the basolateral Na^+^/K^+^-ATPase that maintains an electrochemical gradient, for Na^+^ to enter the cells via an apical channel such as ENaC. This process generates an osmotic gradient across the epithelium that causes water molecules to follow. Water moves either through the transcellular (aquaporins) or paracellular pathway (created with BioRender.com).

**Figure 3 cells-11-01801-f003:**
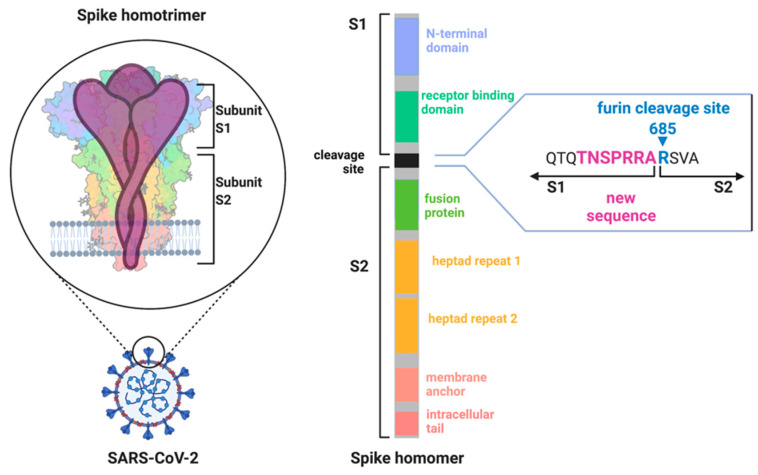
The SARS-CoV-2 spike protein expressed at the virus membrane forms a homotrimer. Each spike subunit can be further divided into two major subdomains (S1 and S2) that are separated by a cleavage site. This cleavage site can be targeted by the protease furin and was identified to include new amino acids (purple) in comparison to SARS-CoV. The inclusion of the new amino acids is considered to make this site more accessible to furin and to be a major reason for the increased transmissibility of SARS-CoV-2 (created by BioRender.com).

## Data Availability

Not applicable.

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
