# Peer review of "COVID-19 and Liquid Homeostasis in the Lung—A Perspective through the Epithelial Sodium Channel (ENaC) Lens"

_cells, 2022, doi:10.3390/cells11111801_

Round 1

Reviewer 1 Report

The manuscript entitled  “COVID-19 and liquid homeostasis in the lung – a perspective 2 through the epithelial sodium channel (ENaC) lens” deals with the discussion of  the implications of SARS-CoV-2 infections from the perspective of impaired function of ENaC focusing on the involvement of  proteases essential for mediating ENaC activity and enabling viral binding to host cell membrane receptors.

In general, I find the manuscript well written and the topic is presented in a comprehensive way.

However, I believe that some aspects have been left out. For example:

  • The role of molecules like camostat mesylate, that is an inhibitor of protease TMPRSS2 affecting both SARS-Cov-2 infection and ENaC activation, should be briefly discussed
  • In the summary the authors stated that a defined approach by increasing protease activity capable of activating ENaC but not the SARS-CoV-2 spike protein is needed. Anyway, is there a protease that is specific only to ENaC, unable to activate SARS-CoV-2 spike protein? The authors should discuss better this statement.

Moreover, I have some suggestions and corrections for the paper :

        The abstract is absent in the final PDF file

Line 38: the sentence that starts with “Some amphibian species” is not clear, maybe the word “that” should be removed.

Line 83: remove full stop after the word “basolateral”

Lines 138-139: please specify in which cellular models this result has been achieved

Line 146:  remove full stop before “With some of the latter”

Line 179: remove full stop before “Particularly”

Line 228: change “if” with “of”

Lines  260-261: please rephrase this sentence, it is not clear

Line 333 : remove “the” before the word “newly”

References 9 and 83 contains a number before the first name. Please fix them.

Author Response

  • The role of molecules like camostat mesylate, that is an inhibitor of protease TMPRSS2 affecting both SARS-Cov-2 infection and ENaC activation, should be briefly discussed

    Response: While we agree that this is an interesting topic authors have consciously decided to not include and discuss any protease inhibitors/antiproteases in the article. Discussing therapeutic strategies including protease inhibitors/antiproteases was not a focus of the article. Initiating a discussion about the effect of camostat mesylate may leave reader wonder why the discussion isn’t covering other relevant drugs and peptides that have been shown to affect proteolytic cleavage of ENaC. It seem that discussing these aspects adequately and in depth may warrant another publication with a specific emphasis on putative therapeutic strategies.
  • In the summary the authors stated that a defined approach by increasing protease activity capable of activating ENaC but not the SARS-CoV-2 spike protein is needed. Anyway, is there a protease that is specific only to ENaC, unable to activate SARS-CoV-2 spike protein? The authors should discuss better this statement.

Response: We have rephrased this aspect in the summary and hope that the statement is now clearer. The statement now reads: “Thus, it might be speculated that improving ENaC function can be achieved by strategies that increase proteolytic cleavage. Improving ENaC cleavage in pulmonary epithelia could therefore be a suitable strategy to treat pulmonary oedema formation in COVID-19 patients with respiratory distress. This however relies on an approach aiming to increasing protease activity of proteases that have a high affinity for ENaC, but are less effective in cleaving the SARS-CoV-2 spike protein. This may include proteases that can effectively cleave the non-furin cleavage site of gamma ENaC but have a low affinity for SARS-CoV-2 spike protein”

Moreover, I have some suggestions and corrections for the paper :

Response: Many thanks for spotting and listing the errors. We have made corrections and changed these things as advised.

The abstract is absent in the final PDF file

Response: Not sure why the abstract did not appear in the pdf file. Will follow up on that.

Line 38: the sentence that starts with “Some amphibian species” is not clear, maybe the word “that” should be removed.

            Response: many thanks for pointing this out. Removed ‘that’ from the sentence.

Line 83: remove full stop after the word “basolateral”

            Response: The full stop has been removed.

Lines 138-139: please specify in which cellular models this result has been achieved

            Response: The cell model used to identify that channel activating protease 1 activates ENaC has been included in the sentence. The statement now reads: “Co-expression of ENaC and CAP1 in Xenopus oocytes resulted in elevated ENaC currents…”.

Line 146:  remove full stop before “With some of the latter”

Response: The full stop has been removed.

Line 179: remove full stop before “Particularly”

Response: The full stop has been removed and the sentence was slightly adapted.

Line 228: change “if” with “of”

            Response: This error has been fixed.

Lines  260-261: please rephrase this sentence, it is not clear

            Response: The sentence has been rephrased. It reads now: “The importance of this site for cell infection and the protease that cleaves the site is known from previous work studying SARS-CoV.”

Line 333 : remove “the” before the word “newly”

Response: the statement has been rephrased. It reads now “…can be explained by the new furin cleavage site.”

References 9 and 83 contains a number before the first name. Please fix them.

Response: Apologies for this error – the numbers have been removed.

Reviewer 2 Report

In the first part, this paper reviews the mechanisms of ENac regulation by proteases, cleaving the subunits at certain specific cleavage sites.

In the second part, the paper highlights similarities between the cleavage sites of ENaC and the cleavage sites of the SARS-CoV2 spike protein. the authors speculate that the same proteases may function not only in activating Enac but also in activating the spike protein. No data, however, is provided to show that the spike protein is indeed cleaved by these proteases, this part remains speculative.

This third part, the speculation is taken one step further by theorizing thata SARS-CoV2 infaction may provide so many spike proteins that no proteases remain available for activating ENaC. Lack of ENaC activation may contribute to the well-known pulmonary edema in COVID-19 pneumonia. 

The second or third part are speculative, the theories are based on published data, and no new data is provided. However, these theories are consistent in itself and may indeed have some merit, which may be confirmed in the future by experimental research.

The language needs some careful editing, since it is impaired by omitted articles, wrong plurals, typos, and doubled words and phrases.

Author Response

Many thanks for the comments. Indeed that article was intended to provide a framework for potential studies based on existing literature. It was not intended to report new findings/data, although we agree that this would be exciting.

We have corrected a number of typos and misspellings and also rephrased some sentences that contained errors and in some cases ambiguous wording.

Hopefully this addresses some of the short comings of the previous draft.